Reducing options of ammonia volatilization and improving nitrogen use efficiency via organic and inorganic amendments in wheat (Triticum aestivum L.)

L. Ramalingappa Pooja 1
Shrivastava Manoj 1
Dhar Shiva 2
Bandyopadhyay Kalikinkar 3
Prasad Shiv 1
Langyan Sapna 4
Tomer Ritu 1
Khandelwal Ashish 1
Darjee Sibananda 1
http://orcid.org/0000-0001-8857-2027 Singh Renu 1 renu_icar@yahoo.com
1 Division of Environment Science, ICAR-Indian Agricultural Reserach Institute , Delhi , India
2 Division of Agronomy, ICAR-Indian Agricultural Reserach Institute , Delhi , India
3 Division of Physics, ICAR-Indian Agricultural Reserach Institute , Delhi , India
4 Division of Germplasm Evaluation, ICAR-National Bureau of Plant Genetic Resources , Delhi , India
Yasin Nasim
Electronic publication date: 2023 Mar 6
Publication date: 2023
Volume: 11
Electronic Location ID: e14965
Received 2022 Sep 9; Accepted 2023 Feb 7
Copyright: © 2023 L Ramalingappa et al.
Copyright year: 2023
Copyright holder: L Ramalingappa et al.
License: This is an open access article distributed under the terms of the Creative Commons Attribution License, which permits unrestricted use, distribution, reproduction and adaptation in any medium and for any purpose provided that it is properly attributed. For attribution, the original author(s), title, publication source (PeerJ) and either DOI or URL of the article must be cited.
License URL: https://creativecommons.org/licenses/by/4.0/

Keywords: Hydroquinone, Calcium carbide, Vesicular arbuscular mycorrhiza, Azotobacter, Garlic powder, Linseed oil, Pongamia oil, Nitrous oxide, Denitrification, N-(n-butyl) thiophosphoric triamide

Funding: The authors received no funding for this work.

==============================
Background

This study investigates the effect of organic and inorganic supplements on the reduction of ammonia (NH3) volatilization, improvement in nitrogen use efficiency (NUE), and wheat yield.

Methods

A field experiment was conducted following a randomized block design with 10 treatments i.e., T1-without nitrogen (control), T2-recommended dose of nitrogen (RDN), T3-(N-(n-butyl) thiophosphoric triamide) (NBPT @ 0.5% w/w of RDN), T4-hydroquinone (HQ @ 0.3% w/w of RDN), T5-calcium carbide (CaC2 @ 1% w/w of RDN), T6-vesicular arbuscular mycorrhiza (VAM @ 10 kg ha−1), T7-(azotobacter @ 50 g kg−1 seeds), T8-(garlic powder @ 0.8% w/w of RDN), T9-(linseed oil @ 0.06% w/w of RDN), T10-(pongamia oil @ 0.06% w/w of RDN).

Results

The highest NH3 volatilization losses were observed in T2 at about 20.4 kg ha−1 per season. Significant reduction in NH3 volatilization losses were observed in T3 by 40%, T4 by 27%, and T8 by 17% when compared to the control treatment. Soil urease activity was found to be decreased in plots receiving amendments, T3, T4, and T5. The highest grain yield was observed in the T7 treated plot with 5.09 t ha−1, and straw yield of 9.44 t ha−1 in T4.

Conclusion

The shifting towards organic amendments is a feasible option to reduce NH3 volatilization from wheat cultivation and improves NUE.

Introduction

Nitrogen (N) in the atmosphere is the principal source of all soil nitrogen. It naturally enters the soil through dead animal and plant residues, biological nitrogen fixation, and chemical N fertilizer applications. Nitrogen fertilizers have become essential to increase crop yield and enhance food quality (Leghari et al., 2016). Food grain crops account for more than 69% of India’s total N fertilizer intake, where wheat alone has a share of 24%. In the crop year 2020–21, India’s wheat production increased from 109 to 118 Mt to meet the growing population’s demand. However, in future, the wheat yield has to be increased by 1.5% per year to satisfy the growing population’s demand (Grain & Feed Annual report, 2021). Ammonia volatilization is a significant cause of nitrogen depletion in agricultural soil worldwide, contributing to low N fertilizer usage, crop production, and indirect nitrous oxide (N2O) emissions. Globally, the average NH3 volatilization risks range from 0.9% to 64% of the applied N (a mean of 17.6%) (Pan et al., 2016). Ammonia volatilization into the atmosphere negatively affects agriculture, ecosystems, and human health.

Further, it also increases the loss of nitrogen for plant growth, thus increasing the cost of cultivation (Brink & Van Grinsven, 2011). Secondary particulate matter (PM10) is formed when NH3 reacts with other air contaminants such as sulfuric acid and nitric acid. It flies long distances and remains in the air for several days, and causes respiratory diseases in humans (Bittman et al., 2014).

A study conducted by Cao et al. (2013) found that the significant loss of N from applied fertilizer was through NH3 volatilization, which was estimated to be 10.0–19.5% of total N loss. Further, they concluded that NH3 volatilization might be the dominant pathway of N loss. Ammonia volatilization favourably occurs in N fertilizers like urea and organic manure. However, ammonia emission is undesirable as it removes N from the soil/plant system and releases it into the atmosphere. Volatilized NH3 is deposited back to the earth’s surface mainly through two processes which are (i) wet deposition through precipitation and (ii) dry deposition when combined with particulate matter. This seriously impacts the environment as it causes acidification of soil and water bodies and eutrophication of the natural ecosystem. It also acts as an indirect source of N2O, a potent greenhouse gas. In some instances, it also directly affects plants under high concentrations and low temperatures. There are many other ways to control N loss, like avoiding the application of urea under high-risk conditions, deep placement, and using controlled-release N-fertilizers. However, the more effective way to control is by using organic and inorganic amendments; some of them are chemical inhibitors like N-(n-butyl) thiophosphoric triamide (NBPT), cyclohexyl phosphoric triamide (CHPT), ammonium thiosulphate (ATS), hydroquinone (HQ), and calcium carbide (CaC2).

Among these, NBPT is used globally, being the most effective in a market that has expanded at 16% per year over the last 10 years. NBPT-treated urea reduces NH3 loss by 53% (Cantarella et al., 2018). The application of 12 kg N ha−1 HQ on an alluvial soil, in conjunction with 120 kg urea-N ha−1, decreased N2O emission by 5% in rice and 7% in wheat systems when compared to the crops grown solely in the presence of 120 kg N ha−1 urea (Modolo et al., 2018).

The slow-N-release coating technology is also suitable for consistent N supply to the plants and reducing loss and contamination effects. We have explored various organic and inorganic amendments to minimize NH3 volatilization losses and enhance NUE and wheat crop productivity. The application of N inhibitors along with urea has increased the average grain yield by 6.8% (Školníková et al., 2022). Nitrification and urease inhibitors are being suggested to decrease N losses and thus increase crop nitrogen usage efficiency (Abalos et al., 2014). Different mechanisms are involved in reducing NH3 volatilization in that NBPT retards the activity of the urease enzyme by competitive inhibition (Fan et al., 2018). Hydroquinone and calcium carbide is also effective in reducing urease activity. Applying HQ to the soil inhibited urease enzyme activity and inhibited or enhanced the activity of other enzymes like polyphenol oxidase, dehydrogenase, protease and phosphatase (Wołejko et al., 2020). The CaC2 has a negative impact on ammonia-oxidizing bacteria, which are present in the soil, and also reduces the activity of dehydrogenase and nitrate reductase enzymes, thereby reducing the N losses (Mahmood et al., 2014). Pongamia oil (karanjin) works as a nitrification inhibitor by reducing the Nitrosomonas activity without affecting Nitrobacter spp. Activity. The treatment with garlic extract, which is rich in the compound thiosulfinate, works on the mechanism of competitive inhibition of urease enzyme as it contains an organosulphur group (-S(O)-S) which is similar to urea which helps in inhibiting the hydrolysis process (Mathialagan et al., 2017).

The world’s population will surpass 9.7 billion by 2050, posing a significant obstacle to achieving food sustainability. The projected increase in the world’s population demands at least a 70% increase in agriculture production in developed countries and 100% in developing countries (Mahmud et al., 2021). Food security of the country and N-based global warming and environmental degradation are interlinked (Bilal & Aziz, 2022). The use of urease inhibitors in agricultural activities has already been investigated as one of the best methods for ensuring adequate nutritional security (Modolo et al., 2018). N-(n-butyl) thiophosphoric triamide treated urea lowers NH3 loss by approximately 53%. The yield benefits from NBPT application on an average of 6.0% and ranges from 0.8% to 10.2% depending on crop types (Cantarella et al., 2018). The effectiveness of NBPT in reducing NH3 loss is well known, but there is still room for progress to increase the amount of inhibition and hence the retention of NBPT-treated urea in the field. However, these inorganic amendments are not economically viable, and most of them restricted their usage in the research areas due to high costs. Upadhyay, Tewari & Patra (2011) investigated that these chemicals are harmful and inhibit the growth of specific beneficial soil microorganisms that indirectly affects crop growth and development. Therefore, the best option that stands before us is using organic amendments to reduce NH3 losses.

Organic amendments are eco-friendly, economically viable, and, most importantly, biodegradable. Therefore, they act as promising N inhibitors. There are many plant-derived inhibitors like Neem (Azadirachta indica) oil, Pongamia (Pongamia glabra) oil, linseed (Linum usitatissimum) oil, garlic extract, and mint (Mentha spicata) as N amendments in inhibiting N losses. Thiosulfinates (TS) present in fresh garlic extract act as a bio-inhibitor of urease enzyme activity and can be used as a potential urease inhibitor in agriculture. Thiosulfinates have the potential to inhibit the urease enzyme, and it takes 3 h and 30 min to start acting on the urease enzyme after its application (Ramli et al., 2014). Applying coated urea fertilizers such as neem-coated urea and pine oleoresin-coated urea in a vertisol reduced the NH3 volatilization by 27.5% and 41.1%, respectively (Jadon et al., 2018). However, these organic amendments lack commercialization and efficient utilization. Therefore, this study concentrates mainly on stressing the importance and comparative study of these organic and inorganic amendments to decrease ammonia volatilization loss and improve increasing NUE and productivity of wheat (Triticum aestivum L.) crops.

Materials and Methods

Details of the experimental site

A field study was conducted at the research farm of the ICAR-Indian Agricultural Research Institute, New Delhi, located at 280°40′N and 770°12′E, at an altitude of 228.16 m above mean sea level (sub-tropical and semi-arid region), during November 2020 to April 2021. The southwest monsoon contributed about 80% of rainfall, an average of 650 mm annually. The pH of the soil was 8 (±0.1), with a sandy clay loam texture. The initial soil nutrients status, when analyzed before sowing of seeds, were found as follows, the available N was low (157 ± 0.5 kg ha−1), medium level of available P (13.5 ± 0.2 kg ha−1), and the available K (196 ± 0.8 kg ha−1).

Experimental design and management

The experiment was conducted in a randomized block design (RBD) with three replications, and each plot area was 12 m2 (4 m × 3 m). In all treatments recommended dose of nitrogen (RDN) was applied except T1 (control), where no RDN was used. The research was carried out through growing wheat variety (HD 2967) with 10 treatments in respective plots, namely labelled as T1 (control), T2 (RDN @ 150 kg ha−1), T3 (RDN + NBPT (N-(n-butyl) thiophosphoric triamide) @ 0.5% w/w RDN), T4 (RDN +HQ (hydroquinone) @ 0.3% w/w of RDN), T5 (RDN + calcium carbide @ 1% w/w of RDN), T6 (RDN + VAM (vesicular arbuscular mycorrhiza @ 10 kg ha−1), T7 (RDN + azotobacter @ 50 g kg−1 seeds) T8 (RDN + garlic powder @ 0.8% w/w of RDN), T9 (RDN + linseed oil @ 0.06% w/w of RDN), and T10 (RDN + pongamia oil @ 0.06% w/w of RDN). Each dose was finalized based on the current usage of these amendments by firms, which were applied in two splits (one at the time of sowing and the second split application 30 days after sowing).

In order to study the effect of organic and inorganic amendments together, three inorganic amendments, two bio-fertilizers, and three organic amendments were chosen based on popularity and literature review. N-(n-butyl) thiophosphoric triamide (Zanin et al., 2015), hydroquinone (Modolo et al., 2018), calcium carbide (Sakariyawo et al., 2020), Garlic powder (Ramli et al., 2014), Linseed oil, and Pongamia oil (Majumdar, 2008) were mixed with urea (RDN) as per the dosage. Wheat seeds were treated with Azotobater spp. and used for sowing in T7. Vesicular Arbuscular Mycorrhiza (as per ICAR RABI Agro-Advisory for Farmers, https://icar.org.in/content/icar-rabi-agro-advisory-farmers) was applied directly into the soil during sowing with an RDN. The recommended dose of fertilizers (RDF) for wheat was applied as N: P2O5:K2O (150:60:40 kg ha−1). Urea, single super phosphate (SSP), and muriate of potash (MOP) were used as a source of N, P2O5, and K2O, respectively.

Method of soil sample collection and analysis

Fresh soil samples were obtained from the 0–15 cm layer of soil at three separate sites from each treatment using an 8 cm tube auger. Three soil samples were obtained from each treatment during tillering, flowering, grain filling, and physiological maturity of the crop. The total fresh soil samples were 30 in number and air-dried for 7 days, sieved through a 2 mm screen, mixed, and placed in plastic bags for further analysis.

Collection and analysis of NH3

The ammonia volatilization was monitored after fertilizer application for up to 10 days using a forced air draft system method (Bhaskar et al., 2022; Stumpe, Vlek & Lindsay, 1984; Bremner, 1965). The closed chambers measuring 20 cm × 20 cm × 50 cm size made of 6 mm acrylic sheets were placed in the field. The volatilized NH3 gas from the soil surface under different treatments was collected in a 2% Boric acid solution containing a mixed indicator (methyl red and bromocresol green). The air inside the chamber was collected into boric acid traps using a vacuum pump having a flow rate of 3 L min−1. The boric acid traps were changed every 24 h. The volatilized NH3 was determined by the titration of boric acid solution with 0.02 N sulphuric acid, and further calculations were done using the formula given below (Eq. (1)).

(1) Volatilizedammonia(mgpermsq.perday)=A∗0.00028∗1000L∗B

Total nitrogen and NH4+-N and NO3−-N analysis

Total nitrogen content in soil was determined by the Kjeldahl method (Kjeldahl, 1883) during the initial and after harvesting of crop growth. The NH4+-N and NO3−-N were analyzed through steam distillation (Bremner & Keeney, 1965) during all four crop growth stages (tillering, flowering, grain-filling, and physiological maturity). The extract was prepared by taking 10 g soils with 0.25 g activated charcoal and 50 mL KCl solution and kept for shaking (30 min), then filtered with Whatman filter paper 1. From the same KCl extract, 10 mL each was taken in two different distillation flasks, and 100 mL of distilled water was added to each flask. In addition, 1 g Devardas alloy was added for the case of NH4+-N estimation and NO3−-N estimation 1 g magnesium oxide (MgO) was added and distilled separately, and these ions were captured in 20 mL of 2% Boric acid and titrated against 0.02 N sulfuric acid. Further calculations were done using Eqs. (2) and (3) for NH4+-N estimation and NO3−-N estimation, respectively.

(2) ExchangeableammonicalN%insoil=(Vs−Vb)∗S∗0.014∗100W=Z1

ExchangeableNH4+-N(ppm)=Z1∗104

(3) ExchangeablenitrateN%insoil=(Vs−Vb)∗S∗0.014∗100W=Z2

ExchangeableNO3−-N(ppm)=Z2∗104

where,

Vs denotes the volume of H2SO4 available for sample titration.

Vb denotes the volume of H2SO4 needed for blank titration.

S = H2SO4 power,

W = Weight of oven-dried soil used for analysis.

Available phosphorus (P) was analyzed using Olsens’ estimation method (Olsen, 1954). First, available P from the soil sample was extracted using 0.5 N NaHCO3 solution buffer at pH 8.5. Then available P in the extract was measured by an ascorbic acid method using a spectrophotometer. Next, available potassium (K) in the soil was measured using an ammonium acetate method (Hanway & Heidel, 1952), where available K was extracted by shaking with neutral normal ammonium acetate for 5 min, and the K was determined using a flame photometer. Finally, soil organic carbon was measured using Walkley and Black’s rapid titration method (Walkley & Black, 1934).

Analysis of nitrous oxide (N2O) and other significant parameters

Nitrous oxide flux was analyzed using the closed chamber method (Herr et al., 2020). In this method, dark PVC boxes were installed, and the samples were drawn every 24 h in the morning using syringes, evacuated into plastic vials, and analyzed chromatographically. Denitrification losses were estimated by the denitrification enzyme assay method described by Smith & Tiedje (1979).

Soil urease activity was analyzed at the 50% flowering stage, calorimetrically, by Bremner & Douglas (1971) method. The normalized difference vegetation index (NDVI) was measured using a green seeker (handheld crop sensor by Trimble, Westminster, CO, USA) at the 50% flowering stage. Infrared gas analyzer (LI-COR Model LI-6400X7 portable photosynthetic system) (IRGA) was used to measure the photosynthetic rate and stomatal conductance. Soil microbial biomass carbon (MBC) and soil microbial biomass nitrogen (MBN) were determined by the chloroform fumigation–extraction method described by Vance, Brookes & Jenkinson (1987) and Brookes et al. (1985), respectively. The N content in grains and straws was also measured using the Kjeldahl method (Kjeldahl, 1883). After harvesting the crop, yield attributes were calculated from each plot.

Calculation of nitrogen use efficiency in wheat

The nitrogen use efficiency can be defined as the ratio of outputs to inputs of nitrogen (i.e., NUE = N yield/N input).

(4) AgronomicefficiencyofN(AEN)(kgha−1)=GYF−GYNAFN

(5) ProductionefficiencyofN(PEN)(kgkg−1)=GYF−GYNTUN−CUN

(6) ApparentNrecovery(ANR)(%)=TUN−CUNAFN×100

TUN = Total N uptake from the fertilized plots (kg ha−1)

CUN = Total N uptake from unfertilized/control plots (kg ha−1)

AFN = Amount of applied fertilizer N (kg ha−1)

GYF = Grain yield in the fertilized plots (kg ha−1)

GYN = Grain yield in unfertilized/control plots (kg ha−1)

Statistical analysis

The measurements obtained from the experimental work were analyzed by using OPSTAT.01 Software (Sheoran et al., 1998) was used to calculate ANOVA, and means were separated using Duncan’s multiple range test (DMRT) at α = 0.05. The data for each variable was evaluated using variance protocol analysis for a randomized block design, which was checked using the “F” test for statistical significance (Gomez & Gomez, 1984). The standard error of means (SEm) and critical difference (CD) parameters were calculated at a 5% significance level.

Results

Ammonia volatilization losses during the wheat growth period

The addition of organic and inorganic amendments in wheat crop have shown effective controlled of ammonia volatilization losses during entire cropping season as shown in Fig. 1. Results revealed that significant losses of N through NH3 volatilization were observed in the initial 5–6 days after N fertilizer application. The highest N volatilization losses were observed in T2 (20.4 kg ha−1 season−1) and the lowest losses were found in T1 (6.4 kg ha−1 season−1). T3 performed well in reducing the NH3 losses by 40% as compared to T2. NH3 losses in T3, T4 and T8 were recorded 12.4, 14.6, and 16.6 kg ha−1 season−1, respectively (Fig. 2). Duncan’s multiple range test revealed that T3 (NBPT) and T4 (HQ) effectively reduced the NH3 losses. T5 (CaC2) and T8 (garlic powder) showed no significant difference among their means. However, among plant-based amendments, T8 with garlic powder-treated urea showed better results by reducing NH3 volatilization losses by 17% compared to only RDF-treated urea (T2). Other treatments were not so effective in reducing the NH3 losses.

Figure 1 Temporal graph of ammonia flux and effect of organic and inorganic amendments on ammonia flux.

Figure 2 Effect of various organic and inorganic amendments on ammonia volatilization losses from the soil.

Initial and final soil nutrient status

Available nitrogen content in soil after wheat crop harvest was found highest in treatment T7 (i.e., Azotobacter treated plot) which was 195 kg ha−1. Detailed observations on available N content for different treatments is given in Table 1. The highest SOC was observed in T7, treated with Azotobacter spp., i.e., 0.63%. It significantly increased the SOC content compared to the initial SOC (0.42%) before sowing the wheat crop, as shown in Fig. 3. The highest level of available P was observed in T6 treated with VAM, i.e., 30.4 kg ha−1, which is more than double the initial value of available P, as shown in Table 1. The initial level of available K was 196 kg ha−1, while in the final analysis after harvesting the crop, it increased to 229 kg ha−1 in T10, as depicted in Table 1.

Table 1 Effect of different amendments on available nitrogen, phosphorous, potassium and soil organic carbon in soil, N concentration in wheat grains and straw.

(Statistically significant at p = 0.05. Means followed by common alphabets are not significantly different among themselves by DMRT.

Treatments	Available N (kg ha−1) after harvesting of the crop	Available P (kg ha−1) after harvesting of the crop	Available K (kg ha−1) after harvesting of the crop	Nitrogen content in grain (%)	Nitrogen content in straw (%)	Agronomic efficiency of N (kg ha−1)	Production efficiency of N (kg kg−1)	Apparent N recovery (%)	
T-1	141g	17.9g	214c	1.5g	0.32h	0	0	0	
T-2	173def	19.7f	221b	1.91f	0.39g	9.1	25.8	35.2	
T-3	172ef	22.5e	222b	2.09c	0.65a	12.9	19.8	65.2	
T-4	171f	23.7d	222b	2.06cde	0.62b	11.6	18.5	62.8	
T-5	177cde	23.7de	223b	2.02de	0.56c	9.8	19.8	49.4	
T-6	187b	30.4a	228a	1.97b	0.53d	9.1	16.5	55.4	
T-7	195a	27.8b	229a	1.98a	0.63ab	12.5	15.4	80.8	
T-8	180c	26.9cb	229a	1.92b	0.54d	10.1	17.4	57.7	
T-9	181cb	27.58bc	227a	2.02cd	0.41f	9.6	21.5	44.8	
T-10	178cd	26.8c	229a	1.91e	0.44e	10.7	24	44.5	
Statistical significance (F test)	Significant	Significant	Significant	Significant	Significant	–	–	–	
C.D.	5.55	1.58	3.30	0.10	0.02	–	–	–	
SE (m)	1.85	0.53	1.10	0.03	0.01	–	–	–	
SE (d)	2.62	0.75	1.56	0.05	0.01	–	–	–	
C.V.	1.83	3.70	0.85	2.88	2.20	–	–	–	

Figure 3 Effect of various organic and inorganic amendments on soil organic carbon (SOC) in the soil after the harvesting of wheat.

Ammonical N (NH4+-N) and Nitrate (NO3−N) in the soil

The available nitrogen forms, mostly NH4+-N and NO3−-N in the soil, decreased from tillering to the grain filling stage (Fig. 4). The mean quantities of NH4 +-N in the soil at various crop growth stages (tillering, flowering, grain filling, and physiological maturity stage) were 78.6, 65.6, 57.9, and 53.5 kg ha−1, respectively. The mean of NO3−-N in the soil at all four growth stages was 70, 63.4, 56.5, and 51.1 kg ha−1, respectively. During the tillering stage, the highest NH4+-N (108 kg ha−1) was observed in T3, i.e., treatment with NBPT amended urea, and the highest NO3−-N (58.5 kg ha−1) was found in RDN treated plot (T2). In all the stages of plant growth, T3 showed significantly higher values of NH4+-N, which was 47% higher than T2. Duncan Multiple Range Test also showed that the mean of treatments T4, T7, and T8 was also significantly at par for the available NH4+-N content in the soil, and in the case of NO3−-N availability, the mean of treatments T4, T7, and T5 also showed significant difference during all the stages of the wheat crop growth.

Figure 4 Effect of various organic and inorganic amendments on NH4+-N and NO3−-N concentration in soil.

Nitrous oxide flux and denitrification losses from the soil

Nitrous oxide emission is directly related to the amount of N fertilizer available in the soil. The highest cumulative N2O flux was observed in T2 (RDF) plots with 6.52 kg N2O-N ha−1 season−1. There were no significant differences between T3, T4, and T5 in the case of N2O emission reduction (Fig. 3). But all of them were significant when compared with T2. Among plant-based inhibitor treatments, T8, having urea amended with garlic powder, showed effective results. On the other hand, the highest denitrification losses were observed in T2 (3.66 kg-N ha−1) (Fig. 5). N-(n-butyl) thiophosphoric triamide treated urea was comparatively effective by reducing the denitrification losses by 35%.

Figure 5 Effect of various organic and inorganic amendments on nitrous oxide flux and denitrification losses in soil.

Soil microbial biomass carbon and nitrogen in the soil

Soil biomass carbon and soil biomass nitrogen (found within living organisms like fungi and bacteria) were observed at the 50% flowering stage of the crop. The highest MBN content was reported in T7, i.e., Azotobacter spp. treated plot with 37 mg kg−1, followed by T6, having VAM treated plot. The highest MBC (136 mg kg−1) was observed in T7, Azotobacter spp. treated plot was 63.9% more than T2 (Fig. 6).

Figure 6 Effect of various organic and inorganic amendments on soil microbial biomass nitrogen and carbon (MBN and MBC) in soil.

Soil urease enzyme activity in the soil

As shown in Fig. 7, under the 50% flowering stage in T2, i.e., only RDN treated plot showed the highest urease activity (18.9 mg urea g−1 soil h−1) and the lowest urease activity (7.71 mg urea g−1 soil h−1) was observed in T3 treatment containing NBPT which was indicated by the effectiveness of urease inhibitor.

Figure 7 Effect of various organic and inorganic amendments on soil urease activity in wheat field.

Physiological parameters of wheat crop

Adding organic and inorganic amendments to the soil has significantly influenced the physiological parameters. The range of photosynthesis rate varied from 15.7 to 23.8 μ mol m−2 s−1, which was recorded during the 50% flowering stage. The highest photosynthetic rate was observed in T7 with 23.8 μ mol m−2 s−1, followed by T3 and T6 (23.3 and 22.9 μ mol m-2 s-1). The range of chlorophyll content varied from 0.66 to 0.77, recorded during the 50% flowering stage. The highest chlorophyll content was observed in both T4 and T7, with 0.77, followed by T6 and T10, showing 0.76. The highest LAI was observed in T7 with 4.59, followed by T3 and T8, which recorded the same LAI, i.e., 4.16. The lowest reading was recorded in treatment T1, i.e., 1.85. The highest stomatal conductance was observed in T7 with 0.48 m mol m−2 s−1, followed by T3 and T8 (0.47 and 0.43 m mol m−2 s−1), respectively.

Nitrogen uptake by wheat

The data about the nitrogen content analyzed in wheat grain and straw samples after harvest depicted the positive effect of both organic and inorganic amendments. The Azotobacter spp. treated plot (T7) showed 38.7% higher nitrogen content in grains when compared to T2 treatments, as shown in Table 1. The highest nitrogen content in straw was observed in T3 (NBPT), almost double the straw nitrogen content of the T1 (control) plot.

Wheat yield and nitrogen use efficiency

The data obtained in grain and straw yield showed significant improvement in yields due to the addition of organic and inorganic amendments. The highest grain yield was observed in T3, i.e., treatment with NBPT amended urea with 5.09 t ha−1, and straw yield was in T4, i.e., treatment with hydroquinone amended urea (Table 2). However, in the case of straw yield, all the treatment means were not significantly different at p = 0.05. The other yield attributes like biological yield, number of tillers per m−2, number of spikes per m−2, number of grains per spike, and harvest index were not significantly influenced by amendments addition (p = 0.05). However, the test weight of the wheat grain of treatment T8 (39.8 g) was significant among all the treatments.

Table 2 Effect of different amendments on yield attributes and yield of wheat.

Mean of straw yield, biological yield, No. of tillers, spikes, grains per spike and harvest index. (NS—non-significant (P = 0.05) and mean of grain yield and test weight were found S-Significant (P = 0.05) when compared to F-table value. Duncan’s multiple range test (DMRT) for comparison of mean yields and test weight of different amendment treatments (α = 0.05) was conducted. Means of grain yield and test weight of different treatments followed by different alphabets are significantly different among themselves and Means with the same letter are not significantly different).

Treatments	Grain yield
(t ha−1)	Straw yield
(t ha−1)	Biological yield
(t ha−1)	No. of tillers	No. of spikes	No. of grains per spike	Harvest index	Test weight
(g)	
T-1	3.14c	7.95	11.09	340.67	302.67	36.00	28.34	34.82 b	
T-2	4.50ab	8.85	13.35	448.67	423.67	40.00	33.72	39.35a	
T-3	5.09ab	7.82	12.90	410.67	374.00	40.00	39.41	39.53a	
T-4	4.89a	9.44	14.33	483.33	456.00	39.33	34.12	39.07a	
T-5	4.61ab	8.16	12.77	432.33	402.33	40.67	36.09	39.40a	
T-6	4.51b	8.30	12.81	403.67	370.67	40.67	35.21	39.58a	
T-7	5.01ab	8.40	13.42	448.00	419.00	41.33	37.37	39.53a	
T-8	4.65ab	8.42	13.07	448.00	416.00	38.67	35.59	39.76a	
T-9	4.59ab	8.42	13.01	459.00	428.33	40.00	35.29	39.53a	
T-10	4.74ab	8.23	12.97	466.00	430.00	40.00	36.57	39.16a	
Statistical significance	S	NS	NS	NS	NS	NS	NS	S	
C.D.	0.76	N/A	N/A	N/A	N/A	N/A	N/A	1.34	
SE (m)	0.25	0.99	1.05	3.25	33.98	33.40	1.03	0.54	
SE (d)	0.36	1.40	1.49	4.59	48.05	47.24	1.45	0.77	
C.V.	9.59	20.41	14.08	15.86	13.56	14.22	4.49	2.41	

As nitrogen content in the grains has increased on amending urea with organic and inorganic amendments, NUE was calculated in terms of Agronomic Efficiency of N (AEN), Production Efficiency of N (PEN), and Apparent N Recovery (ANR) using Eqs. (4)–(6). The highest AEN was found in T3, with 12.9 kg ha−1, and the lowest was in T2, with 9.05 kg ha−1 (Table 1). The highest PEN was observed in T2 with 25.8 kg kg−1, and the lowest was in T7 (15.4 kg kg−1). The average value of PEN was 17.9 kg kg−1. The highest ARN (81%) was seen in T7, and the lowest was in T2 (35%). The mean value of ANR was observed to be 49.6%.

Discussion

Our experimental findings in the case of NH3 volatilization losses depicted that T3 outperformed all other treatments because of competitive inhibition of urease enzyme activity by NBPT in the soil, which has a significant role in urea hydrolysis process where urea molecule is converted into ammonium ion. Next to NBPT, the other inorganic amendments significantly decreased the NH3 volatilization losses. The allicin (thiosulfinate) naturally present in garlic is a bio-inhibitor of urease activity, as it contains an organosulfur functional group like that of the ureas. This might help allicin to decrease urease activity. This is in agreement with the research study conducted by Mathialagan et al. (2017). These amendments have practical implications in reducing the quantity of urea applied to the field, thereby reducing the N losses, even though many studies have been taken on the case of NBPT.

There is still a need to improve the duration of inhibition of urease activity, the shelf life of urea fertilizer coated with NBPT, and the economic viability of these compounds.

Agriculture alone contributes 80−90% of NH3 emissions globally, mainly through volatilization from livestock and synthetic nitrogen fertilizer (Xu et al., 2019). It acts as a pollutant influencing the biosphere through haze formation and soil acidification. N2O, a potent greenhouse gas, is also of great concern, contributes to global warming, and affects human and environmental health. Li et al. (2015) reported that the Limus® (a new urease inhibitor consisting of 75% NBPT and 25% N-(n-propyl) thiophosphoric triamide (NPPT)), showed an average 83% decrease in NH3 losses during winter wheat season in China. Only the addition of NBPT can reduce NH3 volatilization losses by 61% to 74%, as reported by Lasisi, Akinremi & Kumaragamage (2019).

Affendi, Mansor & Samiri (2020) conducted a study by adding various chemical and natural urease inhibitors to reduce ammonia and nitrous oxide losses from soil. They reported similar findings in a combination of thiosulfate with urea. However, they found NBPT was more effective than the combined use of thiosulfate with urea to reduce NH3 volatilization losses. Similar results were reported by Eduardo et al. (2016), reported a decrease of 31.6% N, which used to be lost if not applied with thiosulfinates, and concluded it was an upcoming natural urease inhibitor.

The final analysis of TNC, available N, P, and K, reported an increase in all these nutrient contents to the initial analysis, which helps the next sown crop in nutrition. T7 treated with Azotobacter spp. performed well in increasing N and K content in the soil, and T6 was treated with VAM to increase P content in the soil. An increase in the N content of the soil could be due to adding these bio-fertilizers as amendments that significantly increased residual N and reduced the N fertilizer application for the next crop, which will be sown in that plots. Vesicular Arbuscular Mycorrhiza improves P mobilization; hence uptake by wheat crops and P content in soil increases. The addition of bio-fertilizers as an amendments source to reduce N losses has improved the soil SOC.

The highest SOC was observed in T7, which was treated with Azotobacter spp. i.e., 0.63%. It has doubled the SOC content compared to the initial SOC before sowing the wheat crop, i.e., 0.42%. A similar effect of these bio-fertilizers in increasing total nitrogen content was observed in a study on bio-fertilizer affecting structural dynamics, function, and network patterns of the sugarcane rhizospheric microbiota by Liu et al. (2021). Kader, Mian & Hoque (2002) also found similar results while working on the effect of Azotobacter spp. Inoculants on wheat yield and nitrogen uptake. This result is supported by Suri et al. (2011) while working on the influence of VAM and applied P on root colonization in wheat.

The available N forms, mostly NH4+-N and NO3−-N, decreased from the tillering stage to the grain-filling stage due to uptake and losses. During the tillering stage, the highest NH4+-N and NO3−-N was observed in T3, i.e., treatment with NBPT amended urea due to a slowdown of urease activity, thereby enhancing the accumulation of NH4+ and NO3− ions, and this increased the plant uptake. The urease activity was slowed down when NBPT-amended urea was applied to the soil, thus, enhancing the accumulation of NH4+ ions and increasing the plant uptake of NH4+ and NO3− ions. During N mineralization from the applied N fertilizers, most of the N remains as NH4+ ions, then rapidly converting into NO3− due to the inhibitory effect of NBPT. That also helps in reducing NO3− leaching. A similar result was found by Dhakar et al. (2015) while working on the impact of nitrification inhibitors and various nitrogen sources on soil nitrogen distribution in Kinnow orchards.

As NH4+ availability was initially low for the conversion into N2O, the losses were low in the initial 7 days due to the application of these inhibitors. The application of urease inhibitors as amendments and urea regulates the concentration of NH4+ and NO3− in the soil, thereby commanding NH3 and N2O emissions (Ding et al., 2015). Denitrification losses depend on soil moisture, soil temperature, and N-ions availability. Even though NBPT has no direct effect on lowering denitrification losses, it was observed that it had influenced the reduction of the emission of N2O (Cassim et al., 2021). The reduction in the availability of NO3− has controlled the processes of denitrification and N2O emission, mainly in the summer. The primary controller of denitrification processes in winter wheat is soil aeration and temperature (Aulakh et al., 2001).

Amendments like bio-fertilisers Azotobacter spp. and VAM positively affect soil biomass carbon and soil biomass nitrogen. These microbes help decompose organic matter and release the essential nutrients for plant uptake, increasing the MBC and MBN of the soil. Similar results were observed by Faujdar (2011) while working on the effect of FYM, bio-fertilizers, and zinc on nutrient transformations, soil properties, and yield of maize, and their residual effect on wheat.

The soil urease enzyme was suppressed by N inhibitors by a competitive inhibition mechanism, thereby reducing or slowing down the enzyme’s activity. Rapid losses in N are mainly due to urea hydrolysis by the urease enzyme, which drastically increases pH and NH4+ ion concentration in the soil (Liu et al., 2018). So, this rise in pH and ion concentration increases NH3 volatilization losses. In the entire growth period of wheat, the urease activity in the soil was always at a peak, especially during tillering stage; as it grew to maturity, its’ activity dropped. Similar results were reported by Fu et al. (2020) while working on the effects of urease and nitrification inhibitors on the soil.

The physiological characteristics of wheat analyzed (photosynthesis rate, chlorophyll content, LAI, and stomatal conductance) were also significantly influenced by organic and inorganic amendments. In this study, photosynthesis rate, chlorophyll content, LAI, and stomatal conductance were slightly increased in T7 treatment with azotobacter because of an increase in nitrogen availability, which positively affects these physiological parameters. Furthermore, followed by Azotobacter treated plot, NBPT treated plot showed significant improvement in all these parameters.

The increase in the N content in wheat straw and grain might be due to improved nitrogen availability due to a reduced N loss and better NUE. As nitrogen is the most limiting nutrient in the growth and development of the wheat crop, the better uptake of the nutrient N will affect plant growth which will help to attain global food security by increasing the wheat yield by 1.5% per year. These results were similar to those obtained by Singh et al. (2018) while working on certain microorganisms like Bacillus spp. and Azotobacter spp. Mukhtar, Bashir & Nawaz (2018) found that microorganisms enhance crop growth by making nutrients available by fixing nitrogen and phosphate solubilization. In addition, azotobacter spp. can produce a few metabolites like phytohormones and exopolysaccharides, which help crops absorb nutrients and develop roots (Hindersah et al., 2020). Both are inorganic amendments that have effectively improved the plant’s nutrient availability, thereby increasing yield. These findings were similar to those of Kumar et al. (2015), who reported an increase in grain and biological yield by 22.6% and 17.4%, respectively. It was also reported by Galindo et al. (2020) that NBPT has some effect on metabolic pathways in decreasing urease enzyme activity hence increasing the NUE.

Furthermore, slowing down nutrient release has improved the period of availability of N and increased the crop’s N uptake (Liu et al., 2020). These positive effects, in turn, enhanced wheat’s growth, development, and yield. However, in reducing NH3 volatilization NBPT, HQ amendments performed much better than other amendments, but these amendments have some adverse effects on soil microorganism activity as per the studies by Upadhyay, Tewari & Patra (2011).

Conclusions

The current study concludes that both organic and inorganic amendments significantly reduce NH3 losses, thus increasing available soil N and enhancing the NUE of the crops. Treatment containing NBPT reduced the losses by 40% compared to only RDN without any amendments. These results may be due to strong inhibition of urease activity by NBPT in the soil. Among plant-based amendments, garlic powder-treated urea showed better results in decreasing NH3 volatilization losses by 17% compared to only RDN-treated urea. These positive effects, in turn, enhanced the growth, development, and yields of wheat. Even though inorganic amendments performed much better than organic amendments in reducing NH3 volatilization, opting for organic amendments gives way to sustainable agriculture. Thus, the focus should be more on organic than inorganic amendments to reduce NH3 emissions from agricultural fields.

Supplemental Information

Supplemental Information 1 Raw data.

Click here for additional data file.

Supplemental Information 2 The scheme of present research where different organic and inorganic amendments were added to the soil. The effect on ammonia volatilization was studied by using Forced draft method and temporal variations were recorded.

Click here for additional data file.

Additional Information and Declarations

Competing Interests

Author Contributions

Data Availability

Sapna Langyan is an Academic Editor for PeerJ.

Pooja L. Ramalingappa conceived and designed the experiments, performed the experiments, analyzed the data, prepared figures and/or tables, authored or reviewed drafts of the article, and approved the final draft.

Manoj Shrivastava conceived and designed the experiments, analyzed the data, authored or reviewed drafts of the article, and approved the final draft.

Shiva Dhar analyzed the data, authored or reviewed drafts of the article, and approved the final draft.

Kalikinkar Bandyopadhyay analyzed the data, authored or reviewed drafts of the article, and approved the final draft.

Shiv Prasad analyzed the data, authored or reviewed drafts of the article, and approved the final draft.

Sapna Langyan analyzed the data, authored or reviewed drafts of the article, and approved the final draft.

Ritu Tomer performed the experiments, analyzed the data, authored or reviewed drafts of the article, and approved the final draft.

Ashish Khandelwal analyzed the data, prepared figures and/or tables, and approved the final draft.

Sibananda Darjee analyzed the data, prepared figures and/or tables, authored or reviewed drafts of the article, and approved the final draft.

Renu Singh conceived and designed the experiments, performed the experiments, analyzed the data, prepared figures and/or tables, authored or reviewed drafts of the article, and approved the final draft.

The following information was supplied regarding data availability:

The raw data is available in the Supplemental Files.

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
