# Peer review of "Reducing options of ammonia volatilization and improving nitrogen use efficiency via organic and inorganic amendments in wheat (Triticum aestivum L.)"

_PeerJ, doi:10.7717/peerj.14965_

## Round 0.1 · original submission · Major Revisions

Dear Authors,

Reviewers have shown concern about different sections of your article. You need to improve the introduction by adding review of the literature section in the Introduction section. The statistical analysis section needs improvement. Moreover, typos and grammatical mistakes need to be addressed.

I hope the manuscript may be accepted after incorporation of suggested major revision by reviewers

Reviewer 1 ·

Basic reporting

I have gone through the review assignment and assessed that the topic is worth studying, the experiments are well designed and the manuscript is carefully written. However, I have noticed some concerns during the review process, which must be addressed before the publication of the study.


I could not notice the review of the literature section in the Introduction section. All I could find is the damage of using chemicals in agriculture. Please, keep in mind your study direction is the use of organic supplements. Therefore, you must include the scientific investigations of other researchers who worked on the organic amendments.
The authors have cited studies older than the last 10 years, which must be avoided, except in the case of special circumstances.
There are grammatical mistakes and typos in the manuscript text. I advise a thorough revision for language quality.
You have not provided the details of the methods adopted Furthermore, the corresponding references you mentioned to study the detailed methods also do not describe the detailed methodology. That's too weird. I advise you to mention the detailed methods in the current manuscript along with the reference citation from where you picked the method.
Line 125: perhaps you intended to write 'fertilizer'
You have to insert a space between the numeric value and the respective unit. For example, '10 g', and not '10g'. Similarly, 25 ℃, 0.8 m, 50 mL, etc. Correct it throughout the manuscript.
The software reference must be provided in the following format. Software Name, Version (Developer's Name, Town, Country). Follow it for all the software you used in the current study.
Kindly, note that the discussion section is not to review the agreements and disagreements of your results with the previously published literature. The main purpose of the discussion section is to discuss the unique findings/ observations of your experiments. I advise you to enlist your key findings in bullet form; and then discuss them one by one in the view of published literature. you can provide (i) reasons for those unique results, (ii) practical implications, (iii) their current status in other's opinions, and (iv) their global impact. etc. But, everything you shall write will be backed by a published study. It will transform each bullet into a separate paragraph.
Try to avoid using old references.
There is no statistical comparison for the data provided in the tables. Correct it.

Experimental design

Experimental design is good.

Validity of the findings

Findings are novel but not statistically analyzed.

Reviewer 2 ·

Basic reporting

The manuscript is well construcetd and meets the scietific standards

Experimental design

Experimental design is appropriate, however, statistical analysis needs improvemnt

Validity of the findings

The findings are interesting and valuable in the field of agricultural sciences

Additional comments

Review Report
During present study, the authors have evaluated the effect of organic and inorganic amendments on NUE, NH3 volatilization, N2O flux, nutrient uptake in grain and yield of wheat plants. The research is interesting and useful in the field of agriculture sciences. However, some devastating shortcomings are found which needs to be addressed to improve the quality of the manuscript. The revisions are suggested below;
1. Elaborate objective of the study
2. What is the novelty of this research work?
3. Describe the concise role of used organic and inorganic amendments regarding reduction of NH3 loss in Introduction section.
4. Page 8, line 117, what was the criteria/basis for selection of the dose of the amendments (NBPT, HQ, Calcium Carbide, Garlic powder, Linseed oil, and Pongamia oil etc)
5. Page 9, line 125, Describe the method of NH3 volatilization determination in detail.
6. Page 9, line 127, elaborate the methodology of NH4 + -N and NO3 analysis
7. Page 9, elaborate the method used for Available Phosphorus and available potassium, ammonium acetate and carbon.
8. Page 10, line 170, add more information pertaining role of other amendments to reduce NH3 loss under the heading "NH3 volatilization losses during the wheat growth period"
9. Page 10, line 193, add some more results under heading "Ammonical N (NH4 + -N) and Nitrate (NO3 óN) in soil"
10. Elaborate the heading of table 1. Mean what are values depicting? Full names of abbreviations???
11. Why is CV high in tables? Lower value of CV show more validity of rsults
12. Significance of results has not determined by applying any statistical analysis, Why? It is better to apply some appropriate test (Significance among means like DMRT) for better understanding of the study…..
13. Remove typo and grammatical errors from the manuscript
14. Improve discussion with the help of relevant and latest literature. Make sure that discussion regarding all parameters has been incorporated……Moreover, add discussion regarding impact of NH3 volatilization and N2O flux towards climate change and global warming scenario…

---

## Round 0.2 · Minor Revisions

Dear Authors,

For a general reader, please write the complete term/ symbol/ formula before using its abbreviation. Afterwards, write the abbreviation of the term concerned. However, do not write an abbreviation at the start of a sentence. Use this practice throughout the manuscript.

Regards

Reviewer 1 ·

Basic reporting

The manuscript has been substantially improved. This study focuses to investigate the effect of organic and inorganic supplements on the reduction of ammonia (NH3) volatilization, improvement in Nitrogen Use Efficiency (NUE), and wheat yield. A field experiment was conducted following Randomized Block Design with 10 treatments i.e., T1-without nitrogen (control), T2-Recommended Dose of Nitrogen (RDN), T3-(N-(n-butyl) thiophosphoric triamide (NBPT @ 0.5% w/w of RDN), T4-Hydroquinone (HQ @ 0.3% w/w of RDN), T5-Calcium carbide (CaC2 @ 1% w/w of RDN), T6-Vesicular Arbuscular Mycorrhiza (VAM @ 10 kg ha-1), T7-(Azotobacter @ 50 g kg-1 seeds), T8-(Garlic powder @ 0.8% w/w of RDN), T9-(Linseed oil @ 0.06% w/w of RDN), T10-(Pongamia oil @ 0.06% w/w of RDN). The highest NH3 volatilization losses were observed in T2 at about 20.4 kg ha-1 per season. Significant reductions in NH3 volatilization losses were observed in T3 by 40%, T4 by 27%, and T8 by 17% when compared to the control treatment. Soil urease activity was found to be decreased in plot receiving amendments, T3, T4, and T5. The highest grain yield was observed in the T7 treated plot with 5.09 t ha-1, and straw yield of 9.44 t ha-1 in T4. The shifting towards inorganic and organic amendments is a feasible option to reduce NH3 volatilization from wheat cultivation and improves NUE. The current study concludes that both organic and inorganic amendments significantly reduce NH3 losses, thus increasing soil available nutrients and enhancing the NUE of the crops. Treatment containing NBPT reduced the losses by 40% as compared to only RDN without any amendments. These results may be due to strong inhibition of urease activity by NBPT in the soil. Among plant-based amendments, garlic powder-treated urea showed better results in decreasing NH3 volatilization losses by 17% compared to only RDN-treated urea. These positive effects, in turn, enhanced the growth, development, and yields of wheat. However, in reducing NH3 volatilization, inorganic amendments performed much better than organic amendments, but inorganic amendments showed some adverse effects on soil microorganism’s activity. Thus, the focus should be more on the organic amendments rather inorganic amendments to the reduction of NH3 emissions from agricultural fields. According to my opinion, the manuscript is in the position to be published in the journal.

Experimental design

The experimental design is good.

Validity of the findings

Findings have been compared using statistical methods.

---

## Round 0.3 · Minor Revisions

Dear Authors,

I am pleased to see that you have improved all sections of your article. However, I may observe some headings starting with an abbreviation. Please make the required corrections and resubmit your manuscript

Regards

Reviewer 1 ·

Basic reporting

The authors have substantially improved the manuscript. My assessment comments about the current draft are positive. I recommend the acceptance of the current form of the manuscript.

Experimental design

The experimental design is okay and in accordance with the recorded observations.

Validity of the findings

The findings of the study are in accordance with the study design and have been statistically analyzed.

Additional comments

I recommend the acceptance of the current version.

Reviewer 2 ·

Basic reporting

The mnuscript is well written and sufficient literature relevant to manuscript has been cited.

Experimental design

The experimental design is valid.

Validity of the findings

The findings are meaningfully replicated. Appropriate statistical analysis has been performed. The findings are well stated

Additional comments

The authors have incorporated the suggested changes and may be accepted for publication

---

## Round 0.4 · Minor Revisions

Authors have improved the manuscript. I recommend publication of this manuscript after addressing the following comment from the Section Editor:

> In general I had no issues with the scientific content, but was concerned with the language structure. Perhaps the manuscript should be further proofread to make the flow and tense more relatable. A partially edited PDF markup accompanies this review; I request the manuscript be given a proofreading before acceptance

---

## Round 0.5 · Minor Revisions

In general, I had no issues with the scientific content but was concerned with the language structure. The manuscript should be further proofread to make the flow and tense more relatable.

---

## Round 0.6 · accepted · Accept

Authors have improved their manuscript. I recommend this manuscript for publication.